# Comparing Models of Lateral Station-Keeping for Pitching Hydrofoils

**DOI:** 10.3390/biomimetics4030051

**Published:** 2019-07-22

**Authors:** Peter Gunnarson, Qiang Zhong, Daniel B. Quinn

**Affiliations:** Department of Mechanical and Aerospace Engineering, University of Virginia, Charlottesville, VA 22904, USA

**Keywords:** swimming, biolocomotion, fish schooling, stability, maneuvering, unsteady aerodynamics

## Abstract

Fish must maneuver laterally to maintain their position in schools or near solid boundaries. Unsteady hydrodynamic models, such as the Theodorsen and Garrick models, predict forces on tethered oscillating hydrofoils aligned with the incoming flow. How well these models predict forces when bio-inspired hydrofoils are free to move laterally or when angled relative to the incoming flow is unclear. We tested the ability of five linear models to predict a small lateral adjustment made by a hydrofoil undergoing biased pitch oscillations. We compared the models to water channel tests in which air bushings gave a rigid pitching hydrofoil lateral freedom. What we found is that even with no fitted coefficients, linear models predict some features of the lateral response, particularly high frequency features like the amplitude and phase of passive heave oscillations. To predict low frequency features of the response, such as overshoot and settling time, we needed a semiempirical model based on tethered force measurements. Our results suggest that fish and fish-inspired vehicles could use linear models for some aspects of lateral station-keeping, but would need nonlinear or semiempirical wake models for more advanced maneuvers.

## 1. Introduction

When swimming near obstacles, such as other fish or solid boundaries, fish have to maneuver laterally (left and right). Fish schools can have elaborate lattice structures [1], and individuals sometimes maintain a specific position in a school for days at a time [2]. It remains unclear what competing factors determine an individual’s position in a school [3]—particularly whether the motivations are hydrodynamic [1], sensory [4], or behavioral [2]. Whatever the reason, lateral station-keeping is necessary for maintaining a specific position in the school. As for swimming near solid boundaries, mandarin fish [5] and steelhead trout [6] are known to experience energetic savings near sidewalls. Experiments on pitching hydrofoils have confirmed that these savings extend to simple propulsor shapes, and that stable equilibrium altitudes may exist near sidewalls [7]. In light of these observations, fish may use lateral station-keeping to stay at optimal or preferred distances from other fish and solid boundaries.

One challenge of lateral station-keeping is that many fish are inherently unstable in yaw (twist about dorsoventral axis), because they are pushed from behind by their caudal fin [8]. Many are also unstable in roll (twist about anterior-posterior axis), as demonstrated by the fact that they float “belly up” once anesthetized or dead [9]. When used properly, these instabilities can be advantageous, because passive instability is associated with high maneuverability [8]. Indeed, fish produce quick, tight turns with low accelerations, vastly outperforming man-made vehicles [10].

Much of what we know about lateral maneuvering comes from field/lab observations of fish. Ref. [11] separates lateral fish maneuvers into two groups: trimming forces and powered correction forces. Trimming forces are produced with fine fin adjustments that gently steer the incoming quasi-steady flow. Powered correction forces use sudden fin motions to drive unsteady flows and produce rapid maneuvers. Some tradeoffs between these strategies are understood: trimming forces are generally more efficient but are ineffective at low speeds [5], and larger fins produce higher maneuverability forces but increase drag [12]. The hydrodynamics governing these lateral maneuvers are poorly understood, because classic models for oscillating hydrofoils, e.g., Theodorsen [13] and Garrick [14], were derived for laterally-fixed foils actuated with laterally symmetric motions.

Arguably the simplest way to produce lateral motion is to bias the pitch angle of an oscillating propulsor. Pitch bias produces lateral forces for a tethered airfoil [15], making it a popular method for lateral control in underwater bio-inspired vehicles. The nearly linear relationship between pitch bias and lift provides an easy maneuvering strategy from a control standpoint [16], and pitch bias has successfully controlled, for example, salmon-like [17] and tuna-like [12] robots. Given the success of pitch bias, more work needs to be done to understand how existing hydrodynamic models can be generalized to asymmetric kinematics, particularly for untethered propulsors.

To explore how well existing models capture pitch-biased propulsion, we use a combination of theoretical modeling and hydrofoil maneuver experiments. We use a rigid, pitching, pitch-biased hydrofoil as a reduced-order model of a caudal fin, and we use air bushings to give the foil lateral freedom. We chose a rigid hydrofoil so we could directly compare our experiments with five theoretical models: 2D Theodorsen [13], 3D Theodorsen, 3D Theodorsen plus a wake model, a modified version of Garrick [14], and a semiempirical model based on tethered force tests. The five models, in roughly that order, provide increasingly accurate predictions of a hydrofoil’s dynamic response to biased pitch oscillations. The models are especially effective at predicting high frequency lateral motions. Only the semiempirical model predicts low frequency lateral trajectories well (≤10% error), demonstrating that model-based control may need to be augmented by empirical fits when training fish-inspired robots to move laterally.

## 2. Materials and Methods

### 2.1. Water Channel Setup

To investigate lateral maneuvers, we measured the forces and displacements of a pitching hydrofoil in a water channel (Figure 1A). We chose a rigid, rectangular, low aspect-ratio hydrofoil (chord *c* = 10 cm; span *s* = 15 cm) with a NACA 0012 cross section to facilitate comparisons with previous studies [15,18,19,20]. The foil was suspended from a carriage riding on three air bushings (New Way S301901), giving the carriage near-frictionless lateral freedom when the air was on (Figure 1B). Batteries, compressed air tanks, and a wireless transmitter rode with the carriage, causing the hydrofoil to be effectively laterally untethered during maneuver trials. The carriage massed 4.6 kg; in comparison, a tuna with a similar caudal fin area would mass approximately 19 kg [21,22]. Using a lighter carriage helped accentuate lateral motions for this study while keeping a realistic fin size. To generate thrust, a servo motor (Dynamixel MX-64) actuated the hydrofoil via a driveshaft with pitch oscillations about its leading edge. The pitch angle of the hydrofoil was prescribed as sinusoidal flapping motion plus a pitch bias:(1)$$\theta \left(t\right)=\theta_{\mathrm{bias}}\left(t\right)+\alpha \;\sin\left(2\pi ft\right).$$

An encoder (US Digital A2K 4096 CPR) verified that the commanded pitch motions were reproduced to within 1% in amplitude and 0.5% in frequency. Along the driveshaft, we installed a six-axis force/torque sensor (ATI Mini-40, SI-80-4), which recorded streamwise and lateral forces on the hydrofoil ($F_{x}$ and $F_{y}$). To suppress surface waves, an acrylic plate was placed just under the surface of the water, and a slot in the plate allowed the driveshaft to move laterally. We used the setup to perform two sets of experiments: one to measure tethered thrust (air bushings off), and one to test lateral maneuvers (air bushings on). For the lateral maneuver experiments, we used a photoelectric distance sensor (Baumer OADM 20U2472/S14C) mounted to the carriage to record the lateral position y(t) over time.

In this study, pitch bias causes the carriage to move laterally. In a real fish-like robot, pitch bias would also cause the robot to yaw, because the caudal fin would apply a yaw moment about the robot’s center of mass. As a result, the experimental setup best approximates a large fish undergoing small lateral correction maneuvers, where the change in yaw is minimal. More generally, we used an isolated fin with prescribed yaw angles so we could compare how well linear models predict passive lateral motions on isolated fins. Future work could build off of models of untethered, isolated caudal fins when producing more complex models for fish-like robots.

#### 2.1.1. Tethered Hydrofoil Tests

The goal of the tethered hydrofoil tests was to measure time-averaged thrust and lift forces (${\bar{F}}_{T}$ and ${\bar{F}}_{L}$) produced by biased pitching motions. We define “thrust” and “lift” to be forces parallel and perpendicular to the incoming flow. In the tethered tests, $F_{T}=F_{x}$ and $F_{L}=F_{y}$; in the maneuver tests, $F_{T}\neq F_{x}$ and $F_{L}\neq F_{y}$, because the effective incoming flow is not always in the *x* direction. To measure tethered forces, we turned off the air bushings and centered the hydrofoil in the water channel. We measured forces over a range of frequencies (*f* = 0.5–2 Hz, 0.5 Hz increment), pitch amplitudes (α = 0–15${}^{\circ }$, 3${}^{\circ }$ increment), and levels of pitch bias ($\theta_{\mathrm{bias}}$ = 0–15${}^{\circ }$, 3${}^{\circ }$ increment). The tests were conducted at a single flow speed (*u* = 0.25 m/s; Reynolds number based on chord length = 2.8 × $10^{4}$). This study considers high Reynolds number swimming, where inertial forces dominate over viscous forces. The Theodorsen and Garrick models do not apply to low Reynolds number swimming, where molecular forces play a dominant role [23]. For each permutation of pitch kinematics, we recorded streamwise and lateral forces over 20 pitching cycles. To avoid residual and transient wake effects, we included a 10-s wait period between trials, and a 5 cycle warm-up period before data acquisition. To avoid any sensor drift, we re-zeroed the force sensor before each trial using the known drag coefficient of the hydrofoil at rest. Every permutation of kinematics was repeated 5 times in order to estimate variance between trials. For each trial, forces were filtered (moving-average filter; 5ms window), then time-averaged.

The purpose of the time-averaged forces was to inform a semiempirical model of our maneuver results. Of particular importance for the modeling are the dimensionless Strouhal number (St) and reduced frequency (*k*), defined as
(2)$$St\equiv \frac{fa}{u}\;\mathrm{and}\;k\equiv \frac{\pi fc}{u},$$
where *a* is the tip-to-tip trailing edge amplitude (2csinα). The Strouhal number and reduced frequency ranges were approximately 0 to 0.4 and 0 to 2.5 for the tethered tests, which encompasses the range used in the maneuver tests. Forces were nondimensionalized as $C_{L}$ (lift coefficient), $C_{T}$ (thrust coefficient), and $C_{y}$ (lateral force coefficient) using the dynamic pressure and hydrofoil area:(3)$$C_{L}\equiv \frac{F_{L}}{\frac{1}{2}\rho u^{2}sc},\;C_{T}\equiv \frac{F_{T}}{\frac{1}{2}\rho u^{2}sc},\;\mathrm{and}\;C_{y}\equiv \frac{F_{y}}{\frac{1}{2}\rho u^{2}sc}=\frac{m\ddot{y}}{\frac{1}{2}\rho u^{2}sc}$$

In these dimensionless coefficients, *u* is the incoming flow speed. In the maneuver trials, the effective incoming flow speed is $\sqrt{{\dot{y}}^{2}+u^{2}}$. However, for the slow lateral motions we are considering, $\sqrt{{\dot{y}}^{2}+u^{2}}\approx u$, so we will use *u* (water channel speed) when defining dimensionless coefficients.

Tethered thrust tests also provide an estimate of any misalignment in the hydrofoil relative to the servo motor. We aligned the hydrofoil in the streamwise direction using a vertical laser descending from the carriage. Any remaining misalignment causes a small but non-zero time-averaged lateral force when the recorded pitch bias angle is zero. We used a linear regression to estimate the misalignment in pitch bias (<1${}^{\circ }$), which we subtracted from recorded pitch angles when applying them to our theoretical models.

#### 2.1.2. Maneuvering Experiments

The goal of the maneuvering experiments was to test the dynamic response of the hydrofoil as it shifted from one lateral position to another. The hydrofoil used a proportional (P) controller to adjust its pitch bias based on its distance from a target lateral position ($y_{\mathrm{goal}}$):(4)$$\theta_{\mathrm{bias}}\left(t\right)=K_{P}\left(y\left(t\right)-y_{\mathrm{goal}}\right),$$
where $K_{P}$ is the constant controller gain. Based on preliminary tests, we chose $K_{P}=5{\;}^{\circ }$/cm, which produced quick rise times, a small overshoot, and a small steady-state error—features expected for well-tuned controllers. Proportional controllers are not typically optimal, but we chose a simple controller with a predictable response in order to test different theoretical models of the lateral dynamics. With a deeper understanding of what physical terms govern lateral maneuvers, a more optimal controller could be tuned for an actual fish-inspired robot.

For each trial, we started the hydrofoil 6 cm to the left of its target position. We chose a distance that was large enough to show kinematics-dependent lateral dynamics but small enough to avoid effects of the sidewalls based on prior work (wall proximity ≫c; [7]). The maneuver represents a small lateral correction (60% of the chord) that a fish might make when station-keeping in a school or near a boundary. The frame supporting the carriage was leveled to $\pm 0.006^{\circ }$ using stepper motors that raised/lowered its 4 corners. We ensured a repeatable initial position by pressing the carriage up against a physical stop mounted to the wave-suppression plate. Once the carriage was level and in its starting position, we turned on the air with a solenoidal valve and initiated pitching oscillations ($\alpha =12^{\circ }$; St = 0–0.4, 0.1 increment).

The lateral position data from each trial were separated into their high and low frequency components. Over maneuvering timescales (>1/*f*), the hydrofoil slowly equilibrates to its target position (Figure 1D). Over flapping timescales (<1/*f*), the hydrofoil heaves passively at the pitching frequency (Figure 1E). We decomposed the data into slow and fast components so we could independently test how well the theoretical models worked at those two timescales. We isolated the low frequency component using a moving average filter (window with length 1/f) and the high frequency component by subtracting the low frequency component from the raw data.

## 3. Water Channel Results

The tethered hydrofoil tests showed that the lift coefficient of a flapping hydrofoil increases linearly with pitch bias, with a slope that increases with Strouhal number (Figure 2A). To facilitate direct comparison with the theoretical models presented in Section 4, the Vectored Garrick model (dashed lines; Section 4.3) and the 3D Theodorsen model (Section 4.1) are also overlaid in Figure 2. Both models predict the slope well when the airfoil is not flapping (St = 0), but underestimates the slope for all other Strouhal numbers. One would expect the two models and our data to agree when St = 0, because there the Garrick and Theodorsen models are the same as the well-established thin airfoil theory. However, with a flapping airfoil, the Theodorsen model wrongly predicts no increase in lift, and the Garrick model underpredicts the increase in lift.

Unlike lift, the time-averaged thrust decreases with pitch bias nonlinearly (Figure 2B). Differences between the experimental data and our vectored Garrick model (Section 4.3) can be mainly attributed to drag, which is neglected in the Garrick model. For example, our data show that a non-flapping airfoil produces negative thrust, while the vectored Garrick model predicts a thrust of zero (Figure 2B; St=0). As the Strouhal number increases, our data show an initial decrease in thrust (due to drag from the amplitude of pitching motion), then a rapid increase in thrust (due to higher frequency oscillations overcoming drag). In contrast, the Theodorsen model predicts zero net thrust for all Strouhal numbers, and the vectored Garrick model predicts that thrust increases continuously with Strouhal number. Additionally, the Garrick model predicts that the thrust coefficient decreases slowly with the cosine of the pitch bias, resulting in only a 3% decrease over 15 degrees of pitch bias. In contrast, the experimental data show a steep dropoff, with the thrust coefficient decreasing by more than 100% over 15 degrees of pitch bias for most Strouhal numbers. The Garrick model is unable to capture this large decrease in thrust with pitch bias, presumably because of pressure drag and induced drag that dominate at higher angles of attack.

The maneuver experiments showed that both low and high frequency components of the hydrofoil’s response depend on Strouhal number (Figure 3). At all Strouhal numbers, the hydrofoil heaves passively at the pitching frequency and settles into its new equilibrium position after about 5 s. Because we only used proportional control, the equilibrium position was slightly offset from the goal position. In the low frequency content of the response, the overshoot and settling time of the response decrease with increasing Strouhal number (Figure 3A). In the high frequency content of the response, the amplitude and phase decrease with Strouhal number (Figure 3B). To better understand and model these trajectories, we considered five theoretical models of pitching hydrofoils.

## 4. Theoretical Modelling

We used our filtered position data (Figure 3) to test the predictive power of five theoretical models. Each model predicts the lateral force of a maneuvering hydrofoil controlled by a pitch bias, which is then used to generate a prediction for the trajectory of the airfoil. The first three models are variations of the Theodorsen model [13] for pitching/heaving airfoils. The last two models are proposed modifications to the Theodorsen model: one based on the Garrick model [14] and one using our tethered force data (Figure 2). While the classical Theodorsen and Garrick theories are inviscid and highly linearized, we used them as a basis for the modelling to investigate the physical origin of force producing terms such as downwash from the unsteady wake, finite aspect ratio correction terms, and viscous drag. We define the five models in Section 4.1, Section 4.2, Section 4.3, Section 4.4 using the notation in Table 1.

### 4.1. Theodorsen Model

One of the oldest and most well known theories for the lateral forces produced by a flapping airfoil is Theodorsen’s theory for aeroelastic flutter [13]. While the theory was developed to model aeroelastic flutter in airplane wings, it has been applied to fish swimming because it provides a closed-form solution for the forces on oscillating rigid airfoils [24] and so will form the basis of the models in this paper.

Theodorsen’s theory gives the lateral force produced by a flapping airfoil under several key assumptions. First, the theory is constructed using potential flow and so assumes that the flow around the flapping airfoil is 2D, inviscid, and fully attached. There are three other main assumptions: the airfoil is a flat plate, all angles are small, and the wake produced by the flapping airfoil is flat. For the maneuvers in this study, the small angle and attached flow assumptions are reasonable. However, our flow is not 2D, inviscid, nor does it have a flat wake, and we will explore modifications in later sections.

For sinusoidal pitching and heaving motions (θ=αsin(2πft) and $y=h_{0}\sin\left(2\pi ft+\phi \right)$), Theodorsen theory gives the lateral force coefficient of the airfoil ($C_{L}$) as the sum of a coefficient governed by added mass and ($C_{L,\mathrm{am}}$) and one governed by circulatory lift ($C_{L,\mathrm{circ}}$): (5)$$\begin{array}{ccc} C_{L,\mathrm{Theo}} & = & C_{L,\mathrm{am}}+C_{L,\mathrm{circ}} \end{array}$$
(6)=Im−Camc2u2y¨*+uθ˙*+c2θ¨*−Ccircθ*+y˙*u+3cθ˙*4uC*(k),
where $\theta^{*}=\alpha e^{2\pi fti}$, $y^{*}=h_{0}e^{(2\pi ft+\phi )i}$, $C_{\mathrm{am}}$ and $C_{\mathrm{circ}}$ are constants, dots denote time derivatives, Im is the imaginary part function, and $C^{*}\left(k\right)$ is the Theodorsen lift deficiency function. For the classical Theodorsen theory, the constants $C_{\mathrm{am}}$ and $C_{\mathrm{circ}}$ are equal to π and 2π. The lift deficiency function is a transfer function that accounts for the induced velocity of the wake; when the kinematic inputs are sinusoidal, $C^{*}\left(k\right)$ can be expressed as
(7)$$C^{*}\left(k\right)\equiv F+iG,$$
where *F* and *G* are combinations of Bessel functions applied to the reduced frequency: (8)$$F\equiv \frac{J_{1}\left(k\right)\left[J_{1}\left(k\right)+Y_{0}\left(k\right)\right]+Y_{1}\left(k\right)\left[Y_{1}\left(k\right)-J_{0}\left(k\right)\right]}{{\left[J_{1}\left(k\right)+Y_{0}\left(k\right)\right]}^{2}+{\left[Y_{1}\left(k\right)-J_{0}\left(k\right)\right]}^{2}}\;\mathrm{and}\;G\equiv -\frac{Y_{1}\left(k\right)Y_{0}\left(k\right)+J_{1}\left(k\right)J_{0}\left(k\right)}{{\left[J_{1}\left(k\right)+Y_{0}\left(k\right)\right]}^{2}+{\left[Y_{1}\left(k\right)-J_{0}\left(k\right)\right]}^{2}}.$$

For sinusoidal motions, the presence of the wake behind the airfoil reduces the magnitude of the circulatory lift force by $|C^{*}\left(k\right)|=\sqrt{F^{2}+G^{2}}$ and shifts its phase by $\mathrm{Arg}(C^{*}\left(k\right))=\mathrm{Arg}\left(F+iG\right)$. For non-sinusoidal motion, a Fourier transform can be performed to attenuate and shift the phase of each Fourier component of θ and *y*. The effects of the wake can be ignored by setting $C^{*}\left(k\right)$ equal to one. To convert into the water channel frame (*x*-*y*), thrust and lift must be rotated by the angle of the incoming velocity ($\dot{y}/u$):(9)$$C_{y,\mathrm{Theo}}=C_{L,\mathrm{Theo}}\cos\left(\frac{\dot{y}}{u}\right).$$

### 4.2. Theodorsen with Finite Aspect Ratio Corrections

Classical Theodorsen theory assumes 2D potential flow and neglects 3D effects such tip vortices and planform-dependent added mass. To account for 3D effects, correction coefficients can be added to the circulatory and added mass forces. For the circulatory forces, Ref. [25] show that Prandtl’s lifting line theory (which considers the effect of tip vortices on reducing lift of a wing) extends to oscillating airfoils. In effect, the circulatory forces of a flapping airfoil are reduced by the same factor as a static airfoil, 
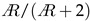
, where 
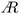
 is the aspect ratio (s/c). For the added mass forces, Ref. [26] found that the added mass of a finite rectangular plate scales approximately with 
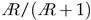
. Following [18], we used these two corrections to modify the constants in Equation (6): (10)
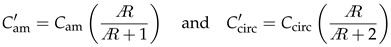


The modified coefficients can be substituted into Equation (Equation 9) to predict the lateral force produced by a finite-span oscillating airfoil:(11)$$C_{y,\mathrm{Theo}3D}={\left.C_{L,\mathrm{Theo}}\cos\left(\frac{\dot{y}}{u}\right)\right|}_{C_{\mathrm{am}}\to C_{\mathrm{am}}^{\prime },C_{\mathrm{circ}}\to C_{\mathrm{circ}}^{\prime }}$$

### 4.3. Theodorsen with Vectored Garrick

The Theodorsen theory was designed to model aeroelastic flutter, so it does not include streamwise forces. This feature precludes Theodorsen from being used to model thrust. It also means Theodorsen cannot explain why pitch bias produces a frequency-dependent lateral force. To verify this conclusion, consider a stationary airfoil with biased pitch oscillations ($\theta \left(t\right)=\theta_{\mathrm{bias}}+\alpha \sin\left(2\pi ft\right)$; $\dot{y}=0$), where the time-averaged lift follows from averaging lift over one flapping cycle:(12)$${\bar{C}}_{L,\mathrm{Theo}}=f\int_{0}^{\frac{1}{f}}C_{L,\mathrm{Theo}}dt=-C_{L,\mathrm{circ}}\theta_{\mathrm{bias}}=-2\pi \theta_{\mathrm{bias}}.$$

This conclusion is true even with finite 
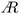
 corrections and even if the small angle assumption of Theodorsen is relaxed by using non-linear added mass forces (see Appendix A). The resulting lift ($2\pi \theta_{\mathrm{bias}}$) is the same lift predicted by thin airfoil theory [27], meaning that Theodorsen predicts no contribution to the time-averaged lift force from the pitch oscillations. In contrast, our experiments show lift force increases with both pitch bias and pitching frequency (Figure 2A).

The Garrick model [14] is a modification to Theodorsen’s theory that includes thrust. Garrick used the same potential flow field as Theodorsen, but integrated the pressure distribution to predict streamwise as well as lateral forces for a flapping airfoil. As an airfoil rotates, the lift is angled by sinθ into the streamwise direction, giving rise to a contribution to the thrust coefficient of $\approx \theta \;C_{L}$ for small θ. At the back of the airfoil, the pressure is assumed to be atmospheric and so does not contribute any net force (Kutta condition). At the front of the airfoil, the flow rushes around the leading edge, the pressure is reduced, and a “leading edge suction force” contributes to thrust. This leading edge suction force points in the direction normal to the leading edge of the airfoil, which for small angle oscillations is approximated as being in the streamwise direction (cosθ≈1). Inviscid computational results have shown that the Garrick theory predicts the time-averaged thrust well, even for finite amplitude pitching motions [19].

The Garrick model predicts that a pitch bias does not alter lateral forces because a small angle approximation is applied to the direction of the leading edge suction force. A pure Garrick model is therefore identical to the Theodorsen model for a static hydrofoil. For this reason, a pure Garrick model is not discussed in this study. Instead we propose a modified version of the Garrick model, a “Vectored Garrick” model, in which we calculate the lift and thrust of the oscillating airfoil using Garrick’s theory and then vector those forces by the effective bias angle.

Because they use the same assumptions, the Garrick and Theodorsen models predict the same lift coefficient ($C_{L,\mathrm{Theo}}$, Equation (6)), but the Garrick model also predicts the thrust coefficient. Using our notation, the Garrick model predicts
(13)$$\frac{F_{T}}{\frac{1}{2}\rho u^{2}sc}=\frac{C_{\mathrm{circ}}}{2}{\hat{\omega }}_{\mathrm{LE}}^{2}+\;C_{L,\mathrm{Theo}}\theta ,$$
where $C_{\mathrm{circ}}$ and $C_{L,\mathrm{Theo}}$ are as defined in Equation (6) and ${\hat{\omega }}_{\mathrm{LE}}$ is the dimensionless vorticity at the leading edge (vorticity scaled with incoming flow speed and the semi-chord). For the special case of sinusoidal pitching and heaving motions (θ=αsin(2πft) and $y=h_{0}\sin\left(2\pi ft+\phi \right)$), Garrick used a round, infinitesimal leading edge to calculate that
(14)ω^LE=2Imθ*+y˙*u+3cθ˙*4uC*(k)−cθ˙*4u.

By integrating over a flapping cycle, Garrick computed the thrust coefficient to be
(15)$${\bar{C}}_{T,\mathrm{Gar}}\equiv \frac{{\bar{F}}_{T}}{\frac{1}{2}\rho u^{2}sc}=\pi k^{2}\left(A_{1}\alpha^{2}+A_{2}\frac{h_{0}^{2}}{c^{2}}+A_{3}\alpha \frac{h_{0}}{c}\right),$$
where
(16)$$A_{1}\equiv \frac{3}{4}C_{\mathrm{am}}+C_{\mathrm{circ}}\left[\left(\frac{1}{k^{2}}+\frac{9}{4}\right)\left(F^{2}+G^{2}-F\right)+\left(\frac{G}{2k}+\frac{3F}{4}\right)\right],$$
(17)$$A_{2}\equiv 4C_{\mathrm{circ}}\left[F^{2}+G^{2}\right],\;\mathrm{and}$$
(18)$$A_{3}\equiv C_{\mathrm{am}}\cos\phi +C_{\mathrm{circ}}\left[2\left(2F+\frac{G}{k}\right)\cos\phi -\frac{2F\sin\phi }{k}+4\left(F^{2}+G^{2}-F\right)\left(\frac{\cos\phi }{2}-\frac{\sin\phi }{k}\right)\right].$$

Because the terms $A_{1},A_{2},$ and $A_{3}$ are nearly constant for the range of reduced frequencies used in this study, the Garrick model for thrust coefficient can be well approximated as a constant times the Strouhal number squared following other thrust models [28].

In order to incorporate thrust into maneuvering with a pitch bias, we propose a model that vectors the thrust from the Garrick model into the direction of the effective pitch bias. This vectoring adds a component of the Garrick-predicted thrust (Equation (Equation 15)) into both the thrust and lift of the hydrofoil: (19)$$C_{L}=C_{L,\mathrm{Theo}}-{\bar{C}}_{T,\mathrm{Gar}}\sin\left(\theta_{\mathrm{bias},\mathrm{eff}}\right)\;\mathrm{and}\;C_{T}={\bar{C}}_{T,\mathrm{Gar}}\cos\left(\theta_{\mathrm{bias},\mathrm{eff}}\right).$$

We use “effective pitch bias”, because the maneuvering hydrofoil is moving laterally, so the incoming velocity is tilted by the angle $\dot{y}/u$, causing the effective pitch bias to be the pitch bias plus the angle of attack induced by lateral motion:(20)$$\theta_{\mathrm{bias},\mathrm{eff}}=\theta_{\mathrm{bias}}+\frac{\dot{y}}{u}.$$

To convert into the water channel frame, thrust and lift must be rotated by the angle of the incoming velocity ($\dot{y}/u$):(21)$$C_{y,\mathrm{Gar}}=C_{T}\sin\left(\frac{\dot{y}}{u}\right)+C_{L}\cos\left(\frac{\dot{y}}{u}\right),$$
where $C_{T}$ and $C_{L}$ are given in Equation (Equation 19). In this “Vectored Garrick Model”, we include the finite aspect ratio correction, that is, we apply the coefficient corrections in Equation (Equation 10).

### 4.4. Semiempirical Model

The Garrick model provides a theoretical basis for thrust, but it uses potential flow and so neglects viscous drag. Additionally, vectoring the thrust from Garrick is breaking the assumption that the airfoil and its wake are flat and aligned with the streamwise direction. These discrepancies are presumably what cause the Vectored Garrick model to deviate from experiments, especially at high values of pitch bias (Figure 2). An alternate solution is to create a semiempirical model which augments the Theodorsen model with empirical data from tethered hydrofoils.

To construct the model, we used our empirical fits of ${\bar{C}}_{L}$ and ${\bar{C}}_{T}$ (Figure 2A,B), which we notate as ${\bar{C}}_{L,\exp}$ and ${\bar{C}}_{T,\exp}$, to predict time-averaged lift and thrust as a function of pitch bias. To convert into the water channel frame, thrust and lift must be rotated by the angle of the incoming velocity ($\dot{y}/u$) to give the predicted lateral force coefficient:(22)$${\bar{C}}_{y,\exp}={\bar{C}}_{T,\exp}\left(\theta_{\mathrm{bias},\mathrm{eff}}\right)\sin\left(\frac{\dot{y}}{u}\right)+{\bar{C}}_{L,\exp}\left(\theta_{\mathrm{bias},\mathrm{eff}}\right)\cos\left(\frac{\dot{y}}{u}\right),$$
where the effective pitch bias (Equation (Equation 20)) is used to account for the tilted incoming velocity during maneuvers.

We propose a model in which our experimentally-measured time-averaged forces replace the time-averaged forces predicted by the Theodorsen model:(23)$$C_{y,\mathrm{semiemp}}=C_{y,\mathrm{Theo}}-{\bar{C}}_{y,\mathrm{Theo}}+{\bar{C}}_{y,\exp},$$
where $C_{y,\mathrm{Theo}}$ is the total lateral force coefficient predicted by Theodorsen (Equation (6)), ${\bar{C}}_{y,\mathrm{Theo}}$ is the time-averaged lateral force coefficient predicted by Theodorsen (Equation (Equation 12)), and ${\bar{C}}_{y,\exp}$ is the time-averaged lateral force coefficient predicted by our tethered hydrofoil experiments (Equation (Equation 22)).

### 4.5. Testing the Theoretical Models

To generate a prediction for the hydrofoil’s lateral position over time (y(t)), we substituted the prescribed θ(t) (Equation (Equation 4)) into each model, then solved for y(t) numerically (Matlab, ODE45). Initial conditions were based on the experiments (y(0)=6 cm; y’(0)=0). In this study, the lateral velocities are small compared to the forward velocity, so we linearized each model using small angle approximations: $\sin\left(\dot{y}/u\right)\approx \dot{y}/u$ and $\cos(\dot{y}/u)\approx 1$. Additionally, the pitch bias angle is small, so for testing the maneuvering prediction, we used the small angle approximation for all expressions involving $\sin(\theta_{\mathrm{bias}})$ and $\cos(\theta_{\mathrm{bias}})$. Because all of the models were linearized, the predicted y(t) is the sum of the high and low frequency components of the predicted y(t). To distinguish between a model’s ability to predict high and low frequency motions, we first used θ(t)=sin(2πft) and compared the predicted response to the measured high frequency response (e.g., Figure 1E), then used $\theta \left(t\right)=\theta_{\mathrm{bias}}\left(t\right)$ and compared the predicted response to the measured low frequency response (e.g., Figure 1D). When testing the models, the effect of the wake was removed by setting *F* equal to 1 and *G* equal to zero, and either the 2D or 3D coefficients $C_{\mathrm{circ}}$ and $C_{\mathrm{am}}$ were used to ignore or include 3D effects. In models 1–4, we performed no empirical fits; all predictions were based on the closed-form coefficients given for each model. Details about implementing the models at high and low frequency are given below.

#### 4.5.1. High Frequency Predictions

We obtained the high frequency component of the predicted response by substituting θ(t)=αsin(2πft) into each model and solving for y(t). Because the Theodorsen model is linear, and θ(t) is sinusoidal here, the solution for y(t) is also sinusoidal and has a closed form solution (see Appendix B for derivation):(24)$$y\left(t\right)=h_{0}\sin\left(2\pi ft+\phi \right),$$
where the heaving amplitude and phase ($h_{0}$ and ϕ) are
(25)$$h_{0}=\frac{\alpha c}{k}\sqrt{\frac{A_{4}^{2}+A_{5}^{2}}{A_{6}^{2}+A_{7}^{2}}}\;\mathrm{and}\;\phi =\mathrm{Arg}\left(\frac{A_{4}+iA_{5}}{A_{6}+iA_{7}}\right),$$
with placeholder constants $A_{4}$–$A_{7}$ defined as
(26)$$\begin{array}{c} A_{4}\equiv C_{\mathrm{circ}}\left[F-\frac{3}{2}kG\right]-C_{\mathrm{am}}k^{2} \end{array}$$
(27)$$\begin{array}{cc} & A_{5}\equiv C_{\mathrm{circ}}\left[G+\frac{3}{2}kF\right]+C_{\mathrm{am}}k \end{array}$$
(28)$$\begin{array}{cc} & A_{6}\equiv 2C_{\mathrm{circ}}G+2C_{\mathrm{am}}k+\frac{8mk}{\rho sc^{2}} \end{array}$$
(29)$$\begin{array}{cc} & A_{7}\equiv -2C_{\mathrm{circ}}F. \end{array}$$

To quantify each model’s ability to predict high frequency motions, we compared the predicted amplitude and phase (Figure 1E) with those of our data. We started by fitting a sine wave to the high frequency component of the lateral position (MATLAB, lsqcurvefit). This fit gave the amplitude of the high frequency lateral motion and the phase relative to the beginning of the trial. Another sine wave curve fit was run on the encoder data to measure the phase of the pitch motion. The difference between the phases gave the phase of the the high frequency heaving motion relative to the high frequency pitching motion. The addition of time-averaged forces from the Garrick model or from experimental data does not affect the high frequency component of the solution, so those models do not offer predictions for high frequency motion.

#### 4.5.2. Low Frequency Predictions

We obtained the low frequency component of the predicted response by substituting $\theta \left(t\right)=\theta_{\mathrm{bias}}\left(t\right)=K_{P}\left(y\left(t\right)-y_{\mathrm{goal}}\right)$ into each model’s prediction for $C_{y}$ and solving for y(t). A complication that arises when using a non-periodic θ(t) is that the lift deficiency function, $C^{*}\left(k\right)$, requires modification. The definition of $C^{*}\left(k\right)$ given in Equations (Equation 7) and (Equation 8) is only valid for sinusoidal motions. If $\theta_{\mathrm{bias}}\left(t\right)$ and y(t) were known, the effect of the lift deficiency function could be included by using a Fourier transform to break $\theta_{\mathrm{bias}}\left(t\right)$ and y(t) into sinusoidal components. However, y(t) is not known *a priori*. Therefore, we will follow [29] and approximate the lift deficiency function using a state-space formulation.

Using the state-space formulation involves solving two coupled differential equations. In the equations, the lift deficiency function, $C^{*}\left(k\right)$, is replaced by the variable $\vec{x}$, which represents the non-periodic downwash induced on the airfoil by the wake. The rank of $\vec{x}$ represents the order of the approximation of $C^{*}\left(k\right)$; we chose the second-order Breuker approximation: $\vec{x}=\left[x_{1};x_{2}\right]$ [30]. The coupled differential equations for $\vec{x}\left(t\right)$ and y(t) are
(30)$$\frac{d\vec{x}}{dt}=\frac{2u}{c}\left[\tilde{A}\vec{x}+\tilde{B}C_{\mathrm{circ}}\left(\theta +\frac{\dot{y}}{u}+\frac{3c\dot{\theta }}{4u}\right)\right]$$
and
(31)$$C_{L,\mathrm{Theo}}=-C_{\mathrm{am}}\frac{c}{2u^{2}}\left(\ddot{y}+u\dot{\theta }+\frac{c}{2}\ddot{\theta }\right)-\tilde{D}C_{\mathrm{circ}}\left(\theta +\frac{\dot{y}}{u}+\frac{3c\dot{\theta }}{4u}\right)-\tilde{C}\vec{x},$$
where $\tilde{A}$, $\tilde{B}$, $\tilde{C}$, and $\tilde{D}$ are constants based on the lift deficiency function [29]:(32)$$\tilde{A}=\left[\begin{array}{cc} -0.3414 & -0.01582\\ 1 & 0 \end{array}\right],\;\tilde{B}=\left[\begin{array}{c} 1\\ 0 \end{array}\right],\;\tilde{C}=\left[\begin{array}{cc} 0.09846 & 0.00757 \end{array}\right],\;\mathrm{and}\;\tilde{D}=0.5177.$$

Equation (Equation 30) factors the circulation shedding from the airfoil into the time-varying downwash; Equation (Equation 31) is a modified version of Equation (6). We set $\vec{x}\left(t=0\right)=\left[0;0\right]$ to model the still wake at the beginning of each trial. To find the low frequency solution for each model, the state-space approximation for $C_{L,\mathrm{Theo}}$ in Equation (Equation 31) is substituted into each expression for $C_{y}$ (Equations (Equation 9), (Equation 19) and (Equation 23)).

Additionally, for the Vectored Garrick model, the time-averaged thrust defined in Equation (Equation 15) is a function of both the pitching motion of the airfoil and sinusoidal lateral motion of the airfoil. To include passive sinusoidal lateral motion into the thrust calculation, the high frequency solution above for $h_{0}$ and ϕ in Equation (Equation 25) was substituted back into Equation (Equation 15) to compute the time-averaged thrust.

To quantify each model’s ability to predict low frequency motions, we compared the predicted overshoot and settling time (Figure 1D) with those of our data. First, we measured the steady-state position by taking the average position of the last 10 seconds for each trial. To determine overshoot, we measured the difference between the steady-state position and the lowest recorded *y* position of the hydrofoil. To determine settling time, we measured the time until changes in *y* position no longer exceeded 2% of the total requested maneuver distance (1.2 mm).

#### 4.5.3. Summary of Model Testing

Each of the five models is a form of the Theodorsen model. The time-averaged forces are either left unchanged (models 1–3), replaced with predictions from the Garrick model (model 4), or replaced with experimental predictions (model 5). In models 1–3, the Theodorsen theory is broken down into three models: 2D, 3D, and 3D with a flat wake. The 3D corrections presented here are based on previous experiments [25,26], so experimental results are important for testing their validity. Additionally, including the wake adds considerable complexity, so it is worth testing to see if the extra computation is necessary to produce a model that works well enough for a fish-like robot. Two additional models were built from the Theodorsen theory with the wake and 3D corrections: Theodorsen with a vectored form of the Garrick model (model 4) and Theodorsen with empirical time-averaged forces (model 5). In summary, we tested the predictions from the following five models:2D Theodorsen No WakeHigh frequency solution is from Equation (Equation 24) with F=1 and G=0, and 2D coefficients $C_{\mathrm{am}}$ and $C_{\mathrm{circ}}$ are used.Low frequency solution is from Equation (Equation 9), where $C_{L,\mathrm{Theo}}$ is taken from Equation (Equation 31) with $\tilde{D}=1$ and $\tilde{C}=0$, and 2D coefficients for $C_{\mathrm{am}}$ and $C_{\mathrm{circ}}$ are used.3D Theodorsen No WakeHigh frequency solution is the same as 2D Theodorsen No Wake, except the 3D coefficients $C_{\mathrm{am}}^{\prime }$ and $C_{\mathrm{circ}}^{\prime }$ are used.Low frequency solution is the same as 2D Theodorsen No Wake, except the 3D coefficients $C_{\mathrm{am}}^{\prime }$ and $C_{\mathrm{circ}}^{\prime }$ are used.3D Theodorsen WakeHigh frequency solution is from Equation (Equation 24), where the 3D coefficients $C_{\mathrm{am}}^{\prime }$ and $C_{\mathrm{circ}}^{\prime }$ are used.Low frequency solution is from Equation (Equation 9), where $C_{L,\mathrm{Theo}}$ is taken from Equation (Equation 31), and 3D coefficients $C_{\mathrm{am}}^{\prime }$ and $C_{\mathrm{circ}}^{\prime }$ are used.Vectored GarrickHigh frequency solution is the same as 3D Theodorsen Wake.Low frequency solution is from Equation (Equation 21) using the same $C_{L,\mathrm{Theo}}$ as the 3D Theodorsen Wake model.SemiempiricalHigh frequency solution is the same as 3D Theodorsen Wake.Low frequency solution from Equation (Equation 23) using the same $C_{L,\mathrm{Theo}}$ as the 3D Theodorsen Wake model.

## 5. Model Testing and Discussion

### 5.1. High Frequency Model Predictions

Only models with finite aspect ratio corrections gave reasonable predictions (<40% error) for the passive heave amplitude (Figure 4A). In the high frequency heave response (Equation (Equation 24)), the amplitude is dominated by $\ddot{y}$ and $\ddot{\theta }$ terms, which relate to added mass. Because the 2D model overestimates the added mass of the foil, it results in an overestimated heave amplitude. When a wake is added, the model switches to underpredicting heave amplitude by about 20%. For the reduced frequencies used in this study, the Theodorsen lift deficiency function is roughly constant with a value of 0.5, meaning the wake reduces the lift magnitude by about one half. It appears that in the real flow, the wake affects lift forces less than the flat wake model predicts. While our results offer insights into the relative predictive power of the models, the absolute values of heave (Figure 3B) depend on the mass of a real fish; our results overpredict heave amplitudes for tuna, for example, because tuna with similarly sized caudal fins are about four times heavier than our carriage [21,22].

All models predict the passive heave phase to within 20% of their experimental values, and adding a wake decreases accuracy (Figure 4B). The poor performance of the wake model is consistent with the errors in passive heave amplitude, where the wake model had an exaggerated effect on lift. Behind a real finite aspect ratio foil, the wake forms interconnected vortex loops [31], and fitted coefficients based on wake structures are needed to predict the wake more accurately [18]. Here we show that while flat wakes indeed misrepresent the wake structure, linear models are still able to predict the heave/phase of passive heave to within 30% accuracy or less, which could be sufficient for some control applications.

Modeling the amplitude and phase of passive high frequency heaving is useful for both fish and fish-like robots, particularly when swimming near obstacles. Both theory (Equation (Equation 15)) and experiments [20] predict that the amplitude and phase of heaving affect thrust and efficiency, with the most efficient phase hovering around 90${}^{\circ }$ [15,32,33]. We observed a passive heaving phase that ranged from 80${}^{\circ }$ (St=0.1) to 120${}^{\circ }$ (St=0.4), suggesting that tuning the phase further could produce high propulsive efficiency. Maintaining phase could also be important in fish schooling, because phase offset between neighboring propulsors is known to affect thrust and efficiency [34,35,36]. Low heave amplitudes could be important for maintaining the tight formation swimming seen in schools [1], or for avoiding collisions when seeking out the known propulsive benefits near the substrate [5,6,7]. High frequency wobbling could limit the minimum spacing between fish and obstacles before collisions occur.

Despite being a highly linearized inviscid theory, the 3D Theodorsen model captures some features of the high frequency heave motion, predicting the amplitude within 30% percent and the phase within 20%. The Theodorsen model was developed for small amplitude, high frequency aeroelastic oscillations, so it is expected to capture high frequency motions well. However, the finite aspect ratio corrections are based on thin-airfoil theory, and the wake model is flat, so the accuracy is noteworthy given that no fitted coefficients were used. For low frequency motions, the errors are higher, and models that include time-averaged streamwise and lateral forces are necessary.

### 5.2. Low Frequency Model Predictions

For a non-flapping airfoil (St = 0), the 3D Theodorsen model with a wake captures the low frequency motion well, with errors in the overshoot and settling time less than 15% (Figure 4A,B). In this particular case of a non-flapping airfoil, the prediction from the semiempirical model is similar to the prediction from the Theodorsen model, because the semiempirical model only adds viscous drag to Theodorsen, which is small compared to the lift slope for a lone hydrofoil (Figure 2, St = 0). Drag forces could be considerably larger for a real fish or fish-inspired robot, where a bluff payload may be upstream of the propulsor. At higher speeds, a fish-inspired robot may therefore benefit from using a semiempirical model that accounts for viscous drag.

For all non-zero Strouhal numbers, it appears that Theodorsen alone is unable to capture the low frequency physics of a flapping airfoil maneuvering with pitch bias. Because the low frequency lateral force is dominated by $\dot{y}$ terms, it can be interpreted as a damping term in the low frequency dynamics of lateral position. Therefore, the fact that 2D Theodorsen overpredicts the quasisteady lift (Figure 2A) helps to explain why it underpredicts overshoot and settling time (Figure 4C,D and Figure 5A). Finite aspect ratio corrections decrease the predicted lift and thus decrease the effective damping. As a result, the 3D Theodorsen model comes closer to predicting overshoot and settling time (Figure 4C,D and Figure 5B). When a wake is added, the model predicts too much of a decrease in lift: the damping is underpredicted, and the overshoot and settling time are overpredicted (Figure 4C,D and Figure 5C). The Vectored Garrick model, whose lateral forces are primarily governed by the 3D Theodorsen model, predicts a similarly underdamped response (Figure 4C,D and Figure 5D).

As with the high frequency response, adding a wake reduces lift more than it should based on experiments (Figure 4C,D). The wake is especially important to the low frequency dynamics, because the lift deficiency function varies throughout the trial. When k=0, the lift deficiency function ($C^{*}\left(k\right)$) is approximately 1, and it decreases to 0.5 as *k* increases. Therefore, when interpreting the high frequency motions, $C^{*}$ could be treated as 0.5, but at low frequencies, $C^{*}$ starts at 0.5 (fast initial change in $\theta_{\mathrm{bias}}$), then shifts to 1 during the trial ($\theta_{\mathrm{bias}}$ reaches equilibrium). Errors introduced by treating the wake as a flat sheet are therefore likely to be accentuated in the low frequency content of the response. As with the high frequency motions, accuracy could be improved further with nonlinear models that incorporate the morphing nature of vortex loops in the wake [18,19].

Only the semiempirical model brings low frequency errors down below 20% (Figure 4C,D). Unlike the flat wake model, the semiempirical model does not overemphasize the wake’s effects, so the lateral position dynamics are more properly damped. Of the models considered, the semiempirical one seems to offer the highest accuracy for predicting low frequency lateral motions. The remaining errors in the semiempirical model are likely due to our assumption that time-averaged forces on a fixed foil apply to a foil that heaves passively, whereas those forces are known to differ based on theory [14] and experiments [20]. The time-average forces could be measured with a heaving foil instead of a static foil to increase the accuracy of the semi-empirical model.

Developing accurate models for the low frequency component of the trajectory is essential for guaranteeing fast and precise maneuvering. The settling time is analogous to asking how quickly a fish-like robot could arrive to a position and stay there, and the overshoot indicates how precisely a fish-like robot could maneuver. Knowing the settling time and overshoot is essential for taking advantage of efficiency gains when swimming near obstacles. For example, maintaining a close lattice structure [1] or a close proximity to a planar surface [5,6] would be costly if constant corrections were needed due to poorly tuned lateral dynamics. Poor tuning could even be deadly, because orientation control is critical to escaping predators and catching prey [37].

We chose to focus on sinusoidal motions, but real fish-like robots may use aperiodic motions during unexpected maneuvers or intermittent swimming gaits. Therefore, it may be useful to extend these models to arbitrary aperiodic actuation. All models are based on the Theodorsen model, which can predict forces caused by arbitrary pitching and heaving motions using the state-space formulation of the Theodorsen function (Equation (Equation 31)). The Vectored Garrick model adds time-averaged thrust, which could be generalized beyond sinusoidal motion using Equation (Equation 13) to directly compute a time-resolved thrust coefficient. In that case, the leading edge vorticity (Equation (Equation 14)) could be calculated in the same way as the Theodorsen model using the state-space formulation of the Theodorsen function. The semi-empirical model adds time-averaged forces, which could be augmented by a library of the time-resolved forces produced by a predefined set of aperiodic motions.

## 6. Conclusions

Classical theoretical models such as Theodorsen appear to provide a good basis for modeling the lateral dynamics of pitch-biased hydrofoils. Nevertheless, the high errors under certain conditions (Figure 4) highlight the need for better nonlinear models or semiempirical tuning when controlling lateral movement. The Strouhal number dependence of lift for pitch-biased oscillating hydrofoils (Figure 2A) demonstrates that the linearity of Theodorsen and Garrick breaks down even at relatively modest values of pitch bias (${\bar{C}}_{L,exp}-{\bar{C}}_{L,Theo}≳0.5$ for $\theta_{bias}≳10^{\circ }$). Modeling drag and non-linear wake effects is apparently necessary for differentiating between the powered correction forces and the quasisteady trimming forces used by fish [11]. For high precision maneuvers, fish and fish-inspired robots may therefore need to rely on trained controllers rather than model-based decisions when station-keeping in obstacle-laden environments.

## Figures and Tables

**Figure 1 biomimetics-04-00051-f001:**
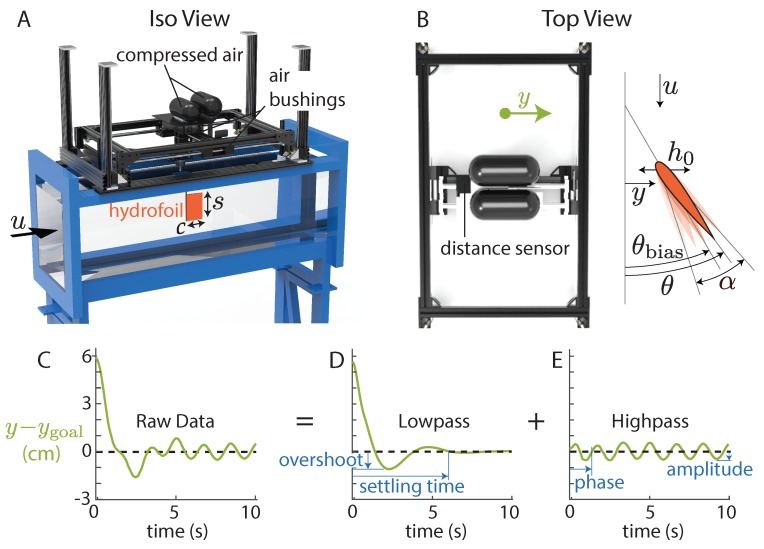
A pitch-biased hydrofoil suspended from air bushings performed lateral maneuvers in a water channel. (**A**) Channel: Rolling Hills 1520; test section: 1.52 m long, 0.38 m wide, and 0.45 m deep. Compressed air at 4500 psi. (**B**) Carriage is free to move laterally (±y). The hydrofoil is driven with pitch oscillations ($\theta =\theta_{\mathrm{bias}}\left(t\right)+\alpha \;\sin\left(2\pi ft\right)$), leading to passive surging (amplitude $h_{0}$). (**C**–**E**) The hydrofoil’s response during a 6 cm lateral maneuver was decoupled into its low (<f) and high (>f) frequency components using a moving average filter.

**Figure 2 biomimetics-04-00051-f002:**
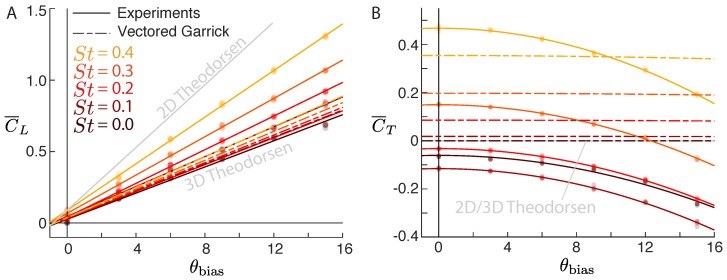
Lateral and streamwise forces deviate from linear theory at high Strouhal numbers and pitch bias angles. (**A**) The time-averaged force perpendicular to the incoming flow (lift) increases linearly with pitch bias ($\theta_{\mathrm{bias}}$) with a slope that increases with Strouhal number. Colored circles show experimental data; solid lines show linear fits (average $R^{2}$ = 0.998). The Vectored Garrick model (dashed lines; Section 4.3) predicts the linear trend but underpredicts the slope. At St=0, the experimental data, the Vectored Garrick model, and the 3D Theodorsen model overlap and so cannot be distinguished on the plot. (**B**) The time-averaged force parallel to the incoming flow (thrust) decreases with pitch bias. Colored circles show experimental data; solid lines show quadratic fits (average $R^{2}$ = 0.994). The inviscid Garrick model always predicts positive thrust and a small decrease with pitch bias of only 3% over 15 degrees of pitch bias. At St=0, the Garrick model is equivalent to the 2D and 3D Theodorsen model.

**Figure 3 biomimetics-04-00051-f003:**
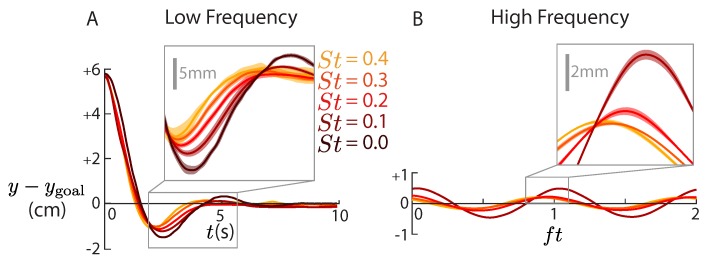
The response in lateral position (*y*) depends on Strouhal number. (**A**) The lowpass-filtered lateral position settles into a new equilibrium after around 5 s. The overshoots and settling times (see Figure 1D) are larger for lower Strouhal numbers. Inset: average, dark lines; average ±σ (n=5), shaded bands. (**B**) The high-pass filtered lateral position oscillates at the pitching frequency. The phases and amplitudes of oscillation (see Figure 1E) decrease with higher Strouhal numbers.

**Figure 4 biomimetics-04-00051-f004:**
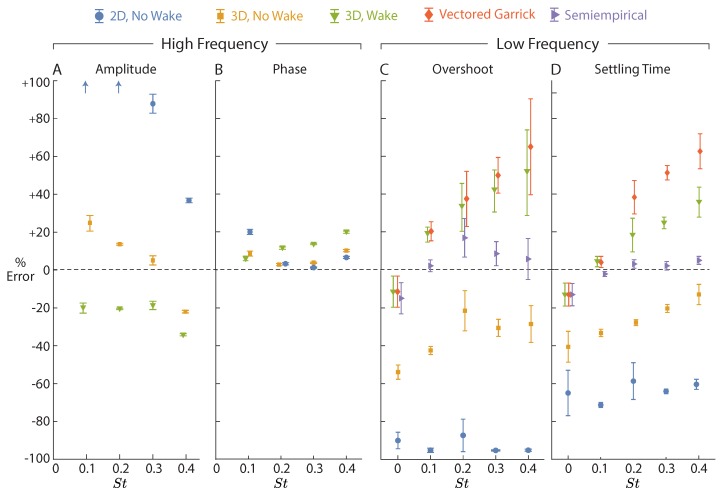
Predictive power varies between models and changes with Strouhal number within each model. Positive % Error implies a metric was overpredicted by the model; negative % Error implies an underprediction. Markers are sometimes shifted left/right to render them distinguishable; all data were taken at exactly St = 0, 0.1, 0.2, 0.3, or 0.4. Errors were calculated based on the high frequency content of the lateral position response (amplitude, **A**; phase, **B**) and the low frequency content of the lateral position response (overshoot, **C**; settling time, **D**). High frequency predictions for the Vectored Garrick and Semiempirical models are the same as the 3D Wake model.

**Figure 5 biomimetics-04-00051-f005:**
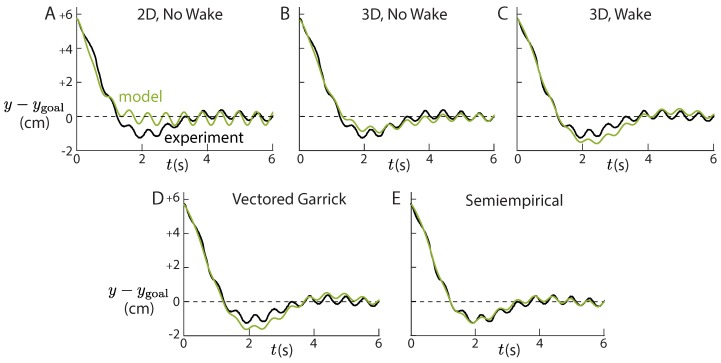
Predictions of lateral position vary between models. The case shown (St = 0.3) demonstrates typical variation between the models. The models which include the wake and 3D corrections (3D, Wake, **C**; Vectored Garrick, **D**; Semiempirical, **E**) capture the high frequency heaving motion better than the models without both the wake and 3D corrections (2D, No Wake, **A**; 3D, No Wake, **B**). Only the semiempirical model accurately predicts both the low and high frequency components.

**Table 1 biomimetics-04-00051-t001:** Variable definitions.

*a*	Tip-to-tip pitching amplitude	*F*	Real part of the Theodorsen function
A˜,B˜,C˜,D˜	Variables for the State-space Theodorsen	FL	Force perpendicular to incoming flow
A1,A2,…	Notation for placeholders	FT	Force parallel to incoming flow
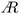	Aspect ratio of the hydrofoil	Fy	Lateral force
*c*	Chord length of the hydrofoil	*G*	Imaginary part of the Theodorsen function
C*(k)	Theodorsen lift deficiency function	h0	Heaving amplitude of the hydrofoil
Cam	Added mass coefficient	J1,J2,Y1,Y2	Bessel functions of the first and second kind
Cam′	Finite 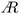 added mass coefficient	*k*	Reduced frequency of the pitching motion
Ccirc	Circulatory force coefficient	Kp	Proportional controller gain
Ccirc′	Finite 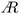 circulatory force coefficient	*m*	Mass of the airfoil and attached rig
CL	Lift coefficient	*s*	Span of the airfoil
CL,am	Added mass component of Theodorsen lift	St	Strouhal number of the pitching motion
CL,circ	Circulatory component of Theodorsen lift	*u*	Incoming flow speed
CL,exp	Empirical lift coefficient	x→	State-space Theodorsen wake downwash
CL,Theo	Theodorsen model lift coefficient	*y*	Lateral position of the airfoil
CT	Thrust coefficient	y*	Complex heaving motion for Theodorsen
CT,exp	Empirical thrust coefficient	α	Angular amplitude of flapping motion
CT,Gar	Garrick model thrust coefficient	θ	Hydrofoil angle relative to water channel
Cy	Lateral force coefficient	θ*	Complex pitching motion for Theodorsen
Cy,exp	Empirical lateral force coefficient	θbias,eff	Pitch bias relative to incoming flow
Cy,Gar	Garrick model lateral force coefficient	θbias	Hydrofoil pitch bias
Cy,semiemp	Semiempirical lateral force coefficient	ρ	Density of water
Cy,Theo	Theodorsen model lateral force coefficient	ϕ	Phase of the heaving relative to pitching
*f*	Flapping frequency (Hz)	ω^LE	Dimensionless leading edge vorticity

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
