# Peer review of "Comparing Models of Lateral Station-Keeping for Pitching Hydrofoils"

_biomimetics, 2019, doi:10.3390/biomimetics4030051_

Round 1
Reviewer 1 Report
I have read the manuscript entitled, “Comparing models of lateral station-keeping for pitching hydrofoils.” In general, I found this to be high-quality work worthy of publication. Most of my comments, while not necessarily minor, should be fairly easily addressed by the authors.
First off, I’m not sure that the title is apt. My take is that station-keeping is about keeping a body on a trajectory or along a prescribed path. The experiments and models in the paper, while similar, seem different - the setup for the experiments is to send the model from point A to point B laterally. This may be a fairly minor nit to pick, so I leave it to the authors discretion. But the title as-is would imply that there is also a level of control in this work to determine how far from a path (or from neighbors in a school) the body/fin currently is.
My biggest comment, that I would like to see the authors respond to, is about the fact that this is an isolated fin. They do address in the manuscript that there may be some differences with a body, that would create more drag and change some of the model parameters. My bigger concern about the “realistic” implementation is that if this fin were attached to the posterior end of a body, it’s biggest effect on that “vehicle” would be to create moment around the center of gravity, yawing the fish instead of shifting it laterally. The modeling done in this paper is important, but I do think there are more dynamics with all 6 axes to be considered before inferring fish motion from caudal fin results.
One other moderate overall comment was the distinction between time-averaged and time-resolved forces. In more realistic applications, especially for something like station-keeping, we wouldn’t assume a purely periodic actuation, so the notion of time-average may be less applicable, and the time-resolved forces may be more important. Can the authors comments here on the applicability of models/controllers based on time-averaged force models?
Some more specific comments:
- line 57: I’m not sure “perturbations” is the right word here, especially in the context of controlling the carriage to move to another location (perturbations would imply an externally applied bump off its track). Would the word “oscillations” work better?
- This is pretty minor, but why are the authors using “l” for chord, when “c” is more common, at least in the fluid mechanics/aerodynamics community?
- Unless I am mistaken, I don’t think the authors included the details on the position measurement in the experiment. “Distance sensor” is noted in figure 1B, and the details of the force transducer, air bushings, pitch encoder, and servo motor are given, but not the device that measures y(t).
- The authors may have a reason for the order of the manuscript as is, but they may want to think about it a little more, or include some more introductory text to section 3. I found it a little confusing to see results of different models in section 3 before they are introduced/explained in section 4.
- Are results of a Garrick model shown at all? I believe that it is clear when the authors are showing or discussing the “vectored Garrick” model in sections 3 and 4, but in section 3, they *seem* to mention a plain Garrick model (before being enhanced to the vectored version). It may all just be vectored Garrick, but I would encourage the authors to make that a little more clear.
- In figure 2B, it’s mentioned on line 133 that “the Garrick model underpredicts how quickly thrust drops off with increasing pitch bias.” From the figure, it looks like it doesn’t drop off at all! I can see a slight decrease if I squint, but the authors may want to be explicit that that model’s thrust is decreasing with bias, but only very slightly. It seems as though a stronger word than “underpredict” is needed here, and may cause confusion otherwise.
- Line 182 - “Inviscid” is misspelled.
- Line 186 - Is there a citation that shows that the Garrick model can be approximated as proportional to St^2? “Because of the relative strengths of these terms” is not enough detail for the point to be delivered in this paper alone. I am not proposing that the authors describe it in more detail here - I anticipate that it’s shown clearly in another work that they can cite.
- Line 215 - the word “can” seems out of place here.
- Line 356 - “offer the highest accuracy for slow pitch control.” I’m not sure what the authors mean here. I think this is the first place a potential distinction between slow pitch and fast pitch is introduced, and I’m not sure why or what the implications are.
- Likewise, at the end of that 2nd to last paragraph, it is stated that a controller could be tuned further by actuating with both heave and pitch when producing empirically fitted forces — I’m not sure what the authors are trying to communicate here either. Is this about running experiments to tune the models? Actuating heave when trying to model lateral motion… I am missing something.
Author Response
We would like to thank the Reviewers for their detailed comments and helpful critiques. We addressed all of their comments, each of which improved and clarified the manuscript. All changes are detailed in point-by-point responses below. All responses and changes to the manuscript are colored in blue.
Reviewer 1
Reviewer: I have read the manuscript entitled, “Comparing models of lateral station-keeping for pitching hydrofoils.” In general, I found this to be high-quality work worthy of publication. Most of my comments, while not necessarily minor, should be fairly easily addressed by the authors.
Response: We thank the Reviewer for this positive summary of our work. We agree with all concerns, and we have addressed each concern below.
First off, I’m not sure that the title is apt. My take is that station-keeping is about keeping a body on a trajectory or along a prescribed path. The experiments and models in the paper, while similar, seem different - the setup for the experiments is to send the model from point A to point B laterally. This may be a fairly minor nit to pick, so I leave it to the authors discretion. But the title as-is would imply that there is also a level of control in this work to determine how far from a path (or from neighbors in a school) the body/fin currently is.
Response: The reason we used “station-keeping” in the title was to emphasize that this study focuses on small lateral correction maneuvers useful in cruise or for staying on a path. The lateral maneuvers we tested were less than one chord length (0.6 c ), the forward velocity (25 cm/s) was much larger than the maximum lateral velocity (~5 cm/s), and the bias angles were small (< 20 degrees). In relation to the water, the hydrofoils in the experiment were moving forward along a slowly drifting path. This situation is unlike rapid tight maneuvering at slow speeds. We therefore chose to keep the title as it is, but we were motivated by this comment to further clarify how our results compare to truly untethered station-keeping (see next response).
My biggest comment, that I would like to see the authors respond to, is about the fact that this is an isolated fin. They do address in the manuscript that there may be some differences with a body, that would create more drag and change some of the model parameters. My bigger concern about the “realistic” implementation is that if this fin were attached to the posterior end of a body, it’s biggest effect on that “vehicle” would be to create moment around the center of gravity, yawing the fish instead of shifting it laterally. The modeling done in this paper is important, but I do think there are more dynamics with all 6 axes to be considered before inferring fish motion from caudal fin results.
Response: Complied. We agree with the Reviewer that it is important to address the added dynamics of a full fish-like robot, particularly yawing motion. As the Reviewer points out, in a truly untethered robot, a lateral force would produce a yawing moment about the center of mass in addition to a lateral acceleration. We added a paragraph to qualify the rationale and implications of studying an isolated fin:
Section 2.1, last paragraph: In this study, pitch bias causes the carriage to move
laterally. In a real fish-like robot, pitch bias would also cause the robot to yaw, because
the caudal fin would apply a yaw moment about the robot’s center of mass. As a result,
the experimental setup best approximates a large fish undergoing small lateral correction maneuvers, where the change in yaw is minimal. More generally, we used an isolated fin with prescribed yaw angles so we could compare how well linear models predict passive lateral motions on isolated fins. Future work could build off of models of untethered, isolated caudal fins when producing more complex models for fish-like robots.
One other moderate overall comment was the distinction between time-averaged and
time-resolved forces. In more realistic applications, especially for something like
station-keeping, we wouldn’t assume a purely periodic actuation, so the notion of time-average may be less applicable, and the time-resolved forces may be more important. Can the authors comments here on the applicability of models/controllers based on time-averaged force models?
Response: Complied. We agree with the Reviewer that understanding non-periodic actuation is important for realistic applications such as rapid maneuvers and intermittent swimming. We chose to focus on sinusoidal motion, but the Theodorsen and Garrick models can be used to predict time-resolved forces for arbitrary motions. Therefore, to explore the applicability of our models when actuation is non-periodic, we added the following text:Section 5.0.2, last paragraph: We chose to focus on sinusoidal motions, but real
fish-like robots may use aperiodic motions during unexpected maneuvers or intermittent
swimming gaits. Therefore, it may be useful to extend these models to arbitrary aperiodic actuation. All models are based on the Theodorsen model, which can predict forces caused by arbitrary pitching and heaving motions using the state-space formulation of the Theodorsen function (Eq. 31). The Vectored Garrick model adds time-averaged thrust, which could be generalized beyond sinusoidal motion using Equation 13 to directly compute a time-resolved thrust coefficient. In that case, the leading edge vorticity (Eq.14) could be calculated in the same way as the Theodorsen model using the state-space formulation of the Theodorsen function. The semi-empirical model adds time-averaged forces, which could be augmented by a library of the time-resolved forces produced by a predefined set of aperiodic motions.
Some more specific comments:
- line 57: I’m not sure “perturbations” is the right word here, especially in the context of
controlling the carriage to move to another location (perturbations would imply an externally applied bump off its track). Would the word “oscillations” work better?
Response: Clarified. In that sentence, we intended to highlight how the semi-empirical model is the only model to accurately predict the low frequency component of y(t). We changed the word“perturbations” to “trajectories” to more accurately convey that we were discussing y(t). We also changed “captured” to “predicted” earlier in the sentence for added clarity.
- This is pretty minor, but why are the authors using “l” for chord, when “c” is more common, at least in the fluid mechanics/aerodynamics community?
Response: We used “l” for the chord to avoid confusion with the coefficients used in the
modelling section (e.g. “C y ”) and because “l” is sometimes used for the length of a fish and for normalization in fish-related studies. However, we agree with the Reviewer that “c” is more common for describing airfoils and would help readers better connect our work with Garrick and Theodorsen theories. We have therefore changed “l” to “c” in the text and in Figure 1 and Table 1.
- Unless I am mistaken, I don’t think the authors included the details on the position
measurement in the experiment. “Distance sensor” is noted in figure 1B, and the details of the force transducer, air bushings, pitch encoder, and servo motor are given, but not the device thatmeasures y(t).
Response: Complied. We thank the Reviewer for catching that we did not fully described the distance sensor. We added a sentence at the end of the first paragraph in section 2.1 which explains that we used a Baumer OADM 20U2472/S14C photoelectric distance sensor to measure y(t).
- The authors may have a reason for the order of the manuscript as is, but they may want to think about it a little more, or include some more introductory text to section 3. I found it a little confusing to see results of different models in section 3 before they are introduced/explained in section 4.
Response: Complied. We agree with the Reviewer that discussing model predictions before models are introduced was confusing. The results of the static force measurements in section 3 are used in section 4 to discuss the limitations of the Garrick and Theodorsen models and to motivate the introduction of the semiempirical model. We therefore chose to include them in section 3. However, as the Reviewer points out, this requires showing models alongside experimental data before the models are defined. To keep the advantages of the section ordering but make the origins of the models clearer, we followed the Reviewer’s advice and added more introductory text at the beginning of section 3:
Section 3, paragraph 1: To facilitate direct comparison with the theoretical models
presented in Section 4, the Vectored Garrick model (dashed lines; Section 4.3) and the
3D Theodorsen model (Section 4.1) are also overlaid in Figure 2.
- Are results of a Garrick model shown at all? I believe that it is clear when the authors are showing or discussing the “vectored Garrick” model in sections 3 and 4, but in section 3, they *seem* to mention a plain Garrick model (before being enhanced to the vectored version). It may all just be vectored Garrick, but I would encourage the authors to make that a little more clear.
Response: Clarified. We only use a vectored version of the Garrick model, and we thank the Reviewer for pointing out that we did not make this clear. We added a paragraph which explains that a plain Garrick model does not predict additional lateral forces, and so only a vectored Garrick model is used in this study:
Section 4.3, paragraph 3: The Garrick model predicts that a pitch bias does not alter
lateral forces because a small angle approximation is applied to the direction of the
leading edge suction force. A pure Garrick model is therefore identical to the Theodorsen model for a static hydrofoil. For this reason, a pure Garrick model is not discussed in this study. Instead we propose a modified version of the Garrick model, a ''Vectored Garrick" model, in which we calculate the lift and thrust of the oscillating airfoil using Garrick's theory and then vector those forces by the effective bias angle.
- In figure 2B, it’s mentioned on line 133 that “the Garrick model underpredicts how quickly thrust drops off with increasing pitch bias.” From the figure, it looks like it doesn’t drop off at all! I can see a slight decrease if I squint, but the authors may want to be explicit that that model’s thrust is decreasing with bias, but only very slightly. It seems as though a stronger word than “underpredict” is needed here, and may cause confusion otherwise.
Response: Clarified. We agree with the Reviewer that it is unclear from Figure 2 that the
Garrick model prediction for C T decreases with pitch bias. The predicted C T drops off with the cosine of the pitch bias, so its decrease is hard to see over the first 16 degrees of pitch bias shown in Figure 2. In the text, we made the language stronger and added a quantitative explanation to more clearly explain the model prediction and how it differs from the experimental data:
Section 3, end of paragraph 2: Additionally, the Garrick model predicts that the thrust
coefficient decreases slowly with the cosine of the pitch bias, resulting in only a 3%
decrease over 15 degrees of pitch bias. In contrast, the experimental data show a steep
dropoff, with the thrust coefficient decreasing by more than 100% over 15 degrees of
pitch bias for most Strouhal numbers. The Garrick model is unable to capture this large
decrease in thrust with pitch bias, presumably because of pressure drag and induced drag that dominate at higher angles of attack.
The description for Figure 2 was also changed to read: “The inviscid Garrick model always predicts positive thrust and a small decrease with pitch bias of only 3% over 15 degrees of pitch bias.”
- Line 182 - “Inviscid” is misspelled.
Response: Complied . We thank the Reviewer for identifying this typo.
- Line 186 - Is there a citation that shows that the Garrick model can be approximated as proportional to St^2? “Because of the relative strengths of these terms” is not enough detail for the point to be delivered in this paper alone. I am not proposing that the authors describe it in more detail here - I anticipate that it’s shown clearly in another work that they can cite.
Response: Complied. We agree that this sentence lacks sufficient detail, and so have included a citation and briefly explained why the Garrick model can be approximated as proportional to St 2 :
Section 4.3, end of paragraph 4: Because the terms A 1, A 2 and A 3 are nearly constant for the range of reduced frequencies used in this study, the Garrick model for thrust coefficient can be well approximated as a constant times the Strouhal number squared, following other thrust models [28]. 28. Smits, A.J. Undulatory and oscillatory swimming. Journal of Fluid Mechanics 2019, 874, P1. doi:10.1017/jfm.2019.284.
- Line 215 - the word “can” seems out of place here.
Response: Complied. We thank the Reviewer for catching this typo, the word “can” has been deleted.
- Line 356 - “offer the highest accuracy for slow pitch control.” I’m not sure what the authors mean here. I think this is the first place a potential distinction between slow pitch and fast pitch is introduced, and I’m not sure why or what the implications are.
Response: Clarified. We agree that the language was imprecise as written. When we wrote “slow pitch control,” we were referring to the low frequency component of the maneuvering experiments. We have changed the sentence to be more clear: “Of the models considered, the semiempirical one offered the highest accuracy for predicting low frequency lateral motions.”
- Likewise, at the end of that 2nd to last paragraph, it is stated that a controller could be tuned further by actuating with both heave and pitch when producing empirically fitted forces — I’m not sure what the authors are trying to communicate here either. Is this about running experiments to tune the models? Actuating heave when trying to model lateral motion… I am missing something.
Response: Clarified. We thank the Reviewer for identifying that this sentence is unclear. We meant to show how the semi-empirical model could be improved by changing our methods for measuring the time-averaged forces. We measured the time-averaged forces of a static foil and used the results in the semi-empirical model for predicting foil maneuvering. However, in the maneuvering experiments, the foil heaves passively, changing the time-averaged forces produced. To match the physics of the maneuvering experiments, the time-average force measurements could be improved by actuating the foil in heave. The last sentence now reads: “The time-average forces could be measured with a heaving foil instead of a static foil to increase the accuracy of the semi-empirical model.”

Reviewer 2 Report
The authors investigated the linear model in predicting the lateral station-keeping of model fish undergoing biased pitch oscillations. The paper is well organized and presented in a clear way. However, there are a few questions need to be answered before I can recommend it for publication in Biomimetics.
1. In the introduction, the authors summarized the swimming in high Reynold’s number, which should be stressed and also compare with low Reynold’s number systems to give readers a general overview on the whole sub branch of active matter. Recommended literatures are: Niu et al. Soft Matter 14 7554-7568 (2018); Wang et al. Acc. Chem. Res.20154871938-1946;
2. A general application range of the five different models is missing or why choose these five models.
3. In Fig. 5, why the high frequency data only have three model predictions?
Author Response
Reviewer: The authors investigated the linear model in predicting the lateral station-keeping of model fish undergoing biased pitch oscillations. The paper is well organized and presented in a clear way. However, there are a few questions need to be answered before I can recommend it
for publication in Biomimetics.
Response: We thank the Reviewer for this positive summary of our work. We agree with all concerns, and we have addressed each concern below.
In the introduction, the authors summarized the swimming in high Reynold’s number, which should be stressed and also compare with low Reynold’s number systems to give readers a general overview on the whole sub branch of active matter. Recommended literatures are: Niu et al. Soft Matter 14 7554-7568 (2018); Wang et al. Acc. Chem. Res.2015487 1938-1946;
Response: We thank the Reviewer for recommending literature to help place our study in a broader context. Our work is only applicable to high Reynolds number swimming, but we agree it is helpful to point readers to the “sub branch” of swimming research that deals with low Reynolds number systems. We therefore chose to reference the general overview of low Reynolds number swimming that the Reviewer suggested (Niu et al. Soft Matter 14 7554-7568) when we introduce the Reynolds number of our experiment:
Section 2.1.1, first paragraph: This study considers high Reynolds number swimming, where inertial forces dominate over viscous forces. The Theodorsen and Garrick models do not apply to low Reynolds number swimming, where molecular forces play a dominant role [23].
23. Niu, R.; Palberg, T. Modular approach to microswimming. Soft Matter 2018, 14, 7554–7568. doi:10.1039/C8SM00995C.
2. A general application range of the five different models is missing or why choose these five models.
Response: We agree with the Reviewer that we can better justify why we chose these models and why they are useful. We added an explanatory sentence to the beginning of section 4:
Section 4, first paragraph: While the classical Theodorsen and Garrick theories are inviscid and highly linearized, we used them as a basis for the modelling to investigate the physical origin of force producing terms such as downwash from the unsteady wake, finite aspect ratio correction terms, and viscous drag.
3. In Fig. 5, why the high frequency data only have three model predictions?
Response: Clarified. While we presented five models in the study, three of the models (3D Wake, Vectored Garrick, and Semiempirical) provide the same prediction for high frequency data. These three models only differ in their low frequency predictions. We added a sentence to the end of the description of Figure 5 for clarity: “High frequency predictions for the Vectored Garrick and Semiempirical models are the same as the 3D, Wake model.” When reviewing the manuscript, we also found the following three minor typos, which have been corrected in our revised manuscript: ● A minus sign was added to Equation 12. ● The arrow showing the highpass amplitude in Figure 1 was slightly adjusted. Previously, the double arrow implied that the highpass amplitude was defined as peak-to-peak, which was incorrect. ● In the last paragraph, a missing } was added to the in-text equation “$\overline{C}_{L,\t{exp} - \overline{C}_{L,\t{Theo}} \gtrsim 0.5$” so that it now correctly reads “$\overline{C}_{L,\t{exp} } - \overline{C}_{L,\t{Theo}} \gtrsim 0.5$.”
